

# Protein language model-based prediction for plant miRNA encoded peptides

Yishan Yue[1], Henghui Fan[2], Jianping Zhao[1] and Junfeng Xia[2]

[1] College of Mathematics and System Science, Xinjiang University, Urumqi, Xinjiang, China
[2] Institutes of Physical Science and Information Technology, Anhui University, Hefei, Anhui, China

## ABSTRACT

Plant miRNA encoded peptides (miPEPs), which are short peptides derived from small open reading frames within primary miRNAs, play a crucial role in regulating diverse plant traits. Plant miPEPs identification is challenging due to limitations in the available number of known miPEPs for training. Existing prediction methods rely on manually encoded features, including miPEPPred-FRL, to infer plant miPEPs. Recent advances in deep learning modeling of protein sequences provide an opportunity to improve the representation of key features, leveraging large datasets of protein sequences. In this study, we propose an accurate prediction model, called pLM4PEP, which integrates ESM2 peptide embedding with machine learning methods. Our model not only demonstrates precise identification capabilities for plant miPEPs, but also achieves remarkable results across diverse datasets that include other bioactive peptides. The source codes, datasets of pLM4PEP are available at https://github.com/xialab-ahu/pLM4PEP.

## INTRODUCTION

MicroRNAs (miRNAs) are short non-coding RNAs that regulate gene expression through a negative regulatory mechanism. They possess the ability to either induce mRNA degradation or inhibit protein synthesis in both plant and animal systems, thus exerting a significant regulatory influence over gene expression (*Carrington & Ambros, 2003*; *Khraiwesh et al., 2010*; *Sun, Lin & Sui, 2019*). A substantial body of research has established the multifaceted roles of miRNAs in a wide array of cellular processes, including those integral to plant growth, development, and stress responses (*Voinnet, 2009*; *Chen, 2009*; *Tang & Chu, 2017*; *Zhang et al., 2020*). The investigation of plant miRNAs is of paramount importance for a comprehensive understanding of the gene regulatory networks functioning in the plants.

Historically, miRNAs were widely regarded as non-coding and thus not translated into peptides. However, recent studies have shed light on their ability to be transcribed by RNA polymerase II into larger primary transcripts known as pri-miRNAs. These plant pri-miRNAs often contain short open reading frame that can encode regulatory peptides, termed miPEPs (*Lauressergues et al., 2015*, *2022*). Interestingly, these miPEPs have been shown to enhance the transcription of the pri-miRNA from which they are derived, in a selective manner. The majority of miRNAs do not function in isolation; instead, they

Corresponding authors
Jianping Zhao, jpzhao@xju.edu.cn
Junfeng Xia, jfxia@ahu.edu.cn

actively engage in intricate regulatory networks to harmonize a variety of developmental processes and stress responses (*Tang & Chu, 2017*). The discovery of miPEPs has opened new avenues for research, offering a novel tool to investigate the specific contributions of individual miRNA families members in the field of plant biology (*Couzigou et al., 2015*).

Bioactive peptides are protein fragments that exhibit positive biological effects and can be produced through enzymatic hydrolysis or fermentation from sustainable food protein sources (*Du et al., 2023*). These bioactive peptides are capable of regulating various physiological functions, leading to the commercial application of certain peptides in pharmaceuticals, nutraceuticals, and cosmetics (*Daliri, Lee & Oh, 2018*; *Du & Li, 2022*; *Fang et al., 2024*; *Tan et al., 2024*). A substantial amount of research has been conducted on a wide range of bioactive peptides, with an increasing volume of experimental data becoming available (*Wei et al., 2019*; *Zhang & Zou, 2020*). However, the ability to identify peptide functionality through experimental methods alone is no longer sufficient to meet current demands. In this context, employing computational methods to guide researchers in the pre-screening of peptide functions has become particularly important (*Chen et al., 2024*; *Goles et al., 2024*; *Fang et al., 2024*). The current research on plant miPEPs is relatively limited, and there is a notable scarcity of computational tools tailored for miPEP identification. This implies that the field of plant miPEPs is still in the early stages and require further exploration. In order to efficiently recognize miPEP, the miPEPPred-FRL (*Li et al., 2023*) predictor has been developed, which employs an advanced feature representation learning framework consisting of feature transformation modules and a cascade architecture. The introduction of miPEPPred-FRL marks a significant step forward, providing a new, less laborious approach for miPEP identification compared with traditional biological experiments. While miPEPPred-FRL demonstrates relatively high accuracy in miPEP identification, there are still challenges that need to be addressed. The current model is constrained by a limited training dataset and a narrow diversity of sample types, which may affect its adaptability to a broader range of plant species and miPEP types, thereby impacting its generalization capabilities. For instance, while miPEPPred-FRL achieves a commendable 90.6% accuracy on independent test sets of Arabidopsis, its performance on legumes and hybrid species is less significant, indicating the need for improvement. In this work, we are dedicated to refining the model to enhance its robustness and generalization, enabling it to perform effectively across diverse datasets. This will facilitate a deeper exploration of miPEPs functionality and mechanisms, providing vital insights to support further research in this field.

In the field of bioinformatics, deep learning-based language models has been increasingly utilized for embedding protein sequence. These models are trained on vast datasets encompassing billions of protein sequences. Protein language models (pLMs) have shown great promise in generating informative protein representations from protein sequence data alone. They offer faster processing times and improved predictive capabilities compared to traditional methods. The embeddings produced by these models have been shown to be highly effective in various downstream tasks, such as prediction of protein structure and function (*Dallago et al., 2021*). Evolutionary scale modeling (ESM2) (*Lin et al., 2023*), released in 2022 by the Fundamental AI Research Protein Team at Meta,

represents the cutting edge of language model for protein sequences. The pLM4ACE (*Du et al., 2024*) is a pioneering model that utilizes ESM2 embeddings for peptide sequence representation in the context of bioactive peptide prediction. The experimental results demonstrate the superiority performance of ESM2 in peptide representation.

In this work, we present a novel bioinformatics tool designed for the accurate identification of miPEPs. Our model was constructed using eight features, including two scales of ESM2 embeddings and six manually encoded features. We evaluated eight machine learning methods as potential classifiers. By comparing the performance of models built with different features and machine learning algorithms, we ultimately obtained the most effective framework. The main contributions of our work are as follows:

(1) Novel ESM2-based peptide classification model. We propose an innovative peptide classification model based on the ESM2 framework. By employing transfer learning, we harness the knowledge learned by the ESM2 model from extensive pre-training on protein sequences and applied to the less explored domain of small-sample plant miPEPs. This strategy significantly enhances the model's recognition performance and generalization, facilitating the accurate identification and classification of miPEPs.

(2) Comparison analysis of ESM2 with different machine learning Models. We conduct an in-depth evaluation and comparison of ESM2's performance when combined with various machine learning models. This analysis provides valuable insights into the most effective machine learning methods peptide embedding in collaboration with ESM2. Our findings offer important guidance for future research aiming to apply ESM2 to peptide feature embedding.

(3) Outstanding model performance across peptide datasets. We evaluate the performance of our model on across a spectrum of bioactive peptide datasets, including those for plant miPEP, neuropeptide, blood-brain barrier peptide, and anti-parasitic peptide. The results demonstrate the superior performance of our model across the evaluated datasets. This compelling evidence substantiates the reliability of our classification model and underscores its potential as a robust and versatile peptide prediction tool, harnessing the capabilities of the ESM2 protein language model.

## MATERIALS AND METHODS

### Datasets

#### Plant miPEPs dataset

We utilized the plant miPEPs dataset constructed by *Li et al. (2023)*, which encompasses a training dataset and three independent testing datasets. The training dataset and independent testing dataset 1 are composed of samples from Arabidopsis thaliana plants, independent testing dataset 2 focused on soybean plants, and independent testing dataset 3 includes hybrid varieties, such as Arabidopsis and cabbage. Table 1 provides the details of these datasets.

**Table 1 Details of the miPEPs dataset.**

| Dataset | Positive | Negative | Total |
|---|---|---|---|
| Training dataset | 250 | 250 | 500 |
| Independent testing dataset 1 | 62 | 545 | 607 |
| Independent testing dataset 2 | 71 | 541 | 612 |
| Independent testing dataset 3 | 23 | 995 | 1,018 |

### Other bioactive peptide datasets

In order to be able to better validate the generalization performance of the model, we selected three other bioactive peptide datasets and keep the testing datasets, the details of which are as follows.

(1) Neuropeptides dataset

We utilized the neuropeptide dataset curated in our previous study (*Bin et al., 2020*), which comprises an equal distribution of 2,425 samples in each positive and negative sample set. The dataset was partitioned into training and test sets in a ratio of 80% and 20% respectively. The final number of positive and negative samples was therefore 485.

(2) Blood-brain barrier peptides dataset

The blood-brain barrier peptide dataset constructed by *Dai et al. (2021)*, with a total of 38 samples in the testing set, including 19 positive samples and 19 negative samples.

(3) Anti-parasitic peptides dataset

The antiparasitic peptide dataset was created by *Zhang et al. (2022)*. The testing set includes 46 positive samples and 46 negative samples. The composition of each bioactive peptide dataset can be found in Table 2.

## Peptide embeddings

### Handcrafted feature encoding

(1) Composition-based feature

This type of feature comprises amino acid composition (AAC) and dipeptide composition (DPC). The AAC is a 20-dimensional vector representing the proportion of the 20 standard amino acids in a specified segment of the peptide sequence. The DPC is a 400-dimensional vector representing the percentage of each possible pair of adjacent residues within the complete peptide sequence.

(2) Physicochemical property-based feature

This category comprises grouped amino acid composition (GAAC), grouped dipeptide composition (GDPC), grouped tripeptide composition (GTPC), and composition, transition, and distribution (CTD). The GAAC feature is a 15-dimensional vector that categorizes the 20 amino acids into five distinct classes based on their physicochemical properties. It quantifies the frequency of each amino acid group's occurrence within the full peptide sequence, the NT5 sequence, and the CT5 sequence. The GDPC is a 25-dimensional vector that records the frequency of dipeptide compositions formed by the five amino acid classes defined in the GAAC feature. The GTPC is a 125-dimensional

**Table 2 Details of the bioactive peptide datasets.**

| Dataset | Positive | Negative | Total |
|---|---|---|---|
| Neuropeptides dataset | 485 | 485 | 970 |
| Blood-brain barrier peptides dataset | 19 | 19 | 38 |
| Anti-parasitic peptides dataset | 46 | 46 | 92 |

vector representing the frequency of tripeptide compositions formed by the five amino acid classes utilized in the GAAC feature. The CTD is a 147-dimensional vector that describes the distribution patterns of amino acids with distinct physicochemical properties within the peptide sequence. It is composed of three components: C (Composition), T (Transition), and D (Distribution). C is represented by a 21-dimensional feature vector, T by a 21-dimensional feature vector, and D by a 105-dimensional feature vector.

### ESM2 feature embedding

ESM2 (*Lin et al., 2023*), a versatile protein language model, provides a broad range of architecture configurations and parameter options to suit a variety of needs. It is designed with flexibility, allowing it to be tailored to align with diverse requirements and computational resources. The model can be configured to run efficiently with a minimal setup of six layers and 8 million parameters, making it a choice for scenarios with constrained computational power or smaller dataset volumes. On the other hand, it can also be expanded to a robust 48 layers and 15 billion parameters, providing a higher capacity to capture subtle patterns and dependencies within extensive and intricate datasets. This scalability enables researchers and practitioners to choose an ESM2 variant that meets their computational efficiency and model complexity needs, achieving an optimal balance. ESM2 is trained on protein sequences derived from the UniRef protein sequence database and has shown remarkable performance across various structure prediction tasks, surpassing other contemporary protein language models (*Lin et al., 2023*).

Given the plant miPEPs dataset's modest sample size and the potential challenge of high dimensionality, our study focused on a comparative analysis of two selected protein language models. The models under consideration are pLM (esm2_t6_8M_UR50D) with 8 million parameters and 320-dimensional output embeddings, and pLM (esm2_t12_35M_UR50D) with 35 million parameters and 480-dimensional output embeddings. After an extensive evaluation, the esm2_t12_35M_UR50D model is chosen for peptide embedding extraction.

### The pLM4PEP framework

The comprehensive overview of pLM4PEP is illustrated in Fig. 1. We leverage the power of ESM2 for representing peptide sequence embeddings, complemented by traditional, manually encoded features. The model is constructed by seamlessly integrating these representations with machine learning methods to enhance its predictive capabilities.

## Performance metrics

Various metrics are employed to evaluate the predictive performance of different models, encompassing widely used measures such as Accuracy (ACC), Matthews correlation coefficient (MCC), and others. The definitions of these metrics are provided below:

$$SP = \frac{TN}{TN + FP} \tag{1}$$

$$PRE = \frac{TP}{TP + FP} \tag{2}$$

$$ACC = \frac{TP + TN}{TP + TN + FP + FN} \tag{3}$$

$$F1 = \frac{2TP}{2TP + FP + FN} \tag{4}$$

$$MCC = \frac{(TP \times TN) - (FP \times FN)}{\sqrt{(TP + FP) \times (TP + FN) \times (TN + FP) \times (TN + FN)}} \tag{5}$$

where TP, TN, FP, and FN represent the number of true positive samples, true negative samples, false positive samples and false negative samples, respectively. In addition, the area under the receiver operating characteristic curve (AUROC) and the area under the precision recall curve (AUPRC) are also considered for model performance evaluation.

## Implementation details

In this work, we use the PyTorch framework to build our prediction model. It is important to note that we use grid search and fivefold cross-validation to optimize the hyperparameters of the algorithms used. To improve the reproducibility and reliability of our model, we have provided the ranges of hyperparameters for the machine learning algorithms in Table S1.

# RESULTS AND DISCUSSION

## Peptide embedding analysis

We constructed predictive models by integrating two variants of ESM2, each with a distinct scale, with six manually encoded feature descriptors. This integration was achieved by employing a combination of eight prevalent machine learning methods: support vector machine (SVM), logistic regression (LR), multilayer perceptron (MLP), Random Forest (RF), K-nearest neighbors (KNN), extreme random trees (ERT), extreme gradient boosting (XGBoost), and adaptive boosting (AdaBoost). The performance of each model was evaluated using the AUROC metric, and the corresponding values obtained on the training set are documented in Table 3. Our observations indicate that the models leveraging ESM2 embeddings outperform those models based on manually encoded features in terms of performance.

In the miPEPPred-FRL (Li et al., 2023) study, a comparison analysis was conducted to evaluate the performance of models using two distinct types of features: those based on protein sequences and those derived from protein language models and deep learning models. In order to facilitate a comprehensive comparison, three protein language models, namely ProteinBERT (Brandes et al., 2022), ProtBert (Elnaggar et al., 2022), and Bert-

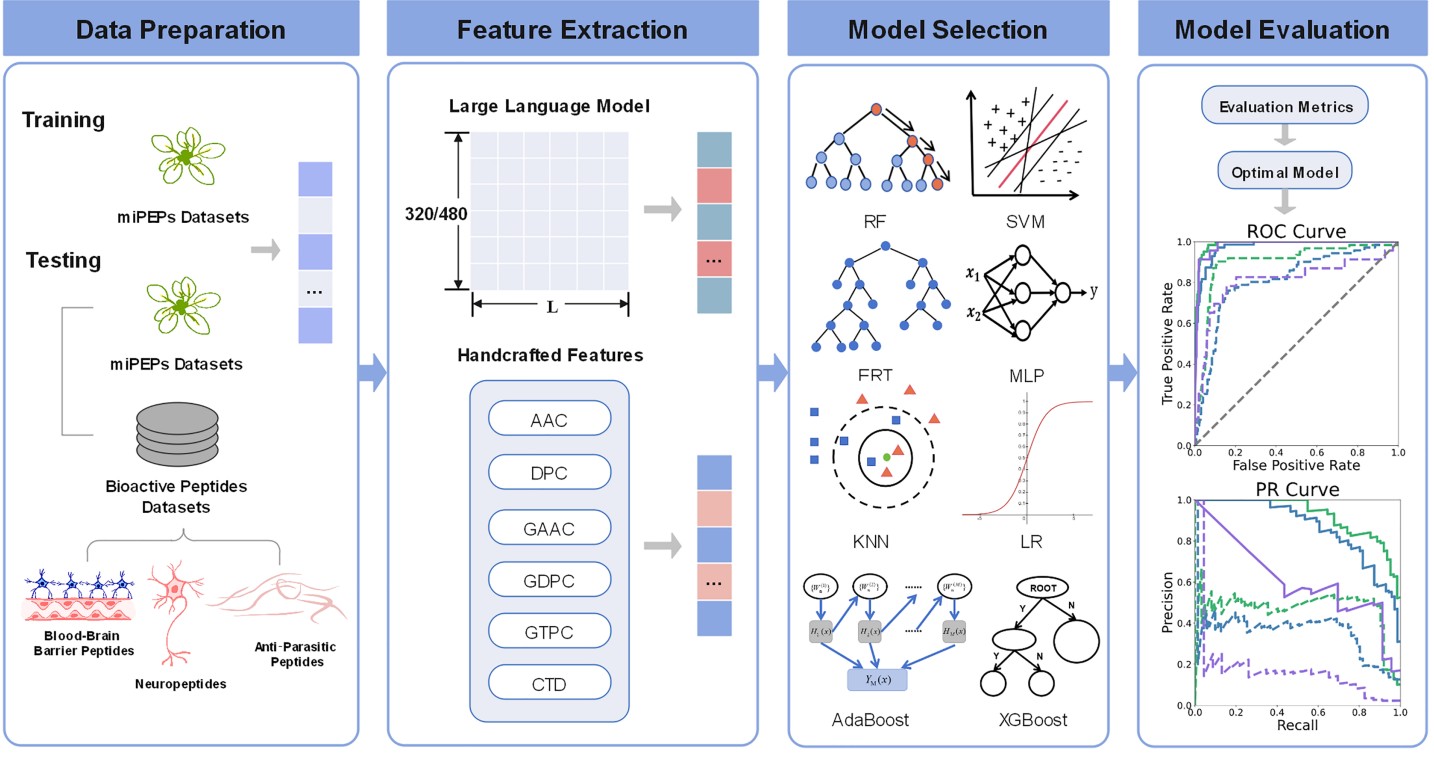

**Figure 1** **The framework of pLM4PEP.** (I) We retrieved peptide sequences from the miPEPs datasets. (II) We extracted crucial features from the peptide sequences, which include a blend of features derived from evolutionary scale modeling (ESM) and those based on handcrafted features. (III) We combined the extracted features with ML algorithms to build models capable of recognizing the miPEPs. (IV) The final stage was dedicated to the thorough evaluation of the constructed models to assess their performance and reliability.

Protein (*Zhang et al., 2021*), as well as three deep learning models, namely CNN, LSTM, and CNN-LSTM, were selected. Their evaluation revealed that protein sequence-based features consistently outperformed the other methods across all machine learning classifiers. In contrast, features derived from protein language models and deep learning models did not match the effectiveness of sequence-based features. One potential explanation for this discrepancy is that features directly extracted from protein language models may not adequately capture the unique characteristics of miPEPs. To demonstrate the superiority of ESM2 and its effectiveness as a feature embedding method, we compared the experimental results of ESM2 with the aforementioned features on four machine learning classifiers. The comparative results are presented in Fig. 2. From Fig. 2, it can be observed that both sizes of ESM2 significantly outperform the sequence-based and pLLM and DLM-based feature extraction methods across various evaluation metrics on the selected four machine learning classifiers.

## Selection of the final classification model

We conducted experiments by combining two variants of ESM2, distinguished by their parameter sizes of 8 million (esm2_t6_8M_UR50D) and 35 million (esm2_t12_35M_UR50D), respectively, with eight machine learning methods to establish

**Table 3  Model performance of various peptide embedding approaches.**

| Feature | LR | AB | XGB | SVM | RF | ERT | KNN | MLP |
|---------|------|------|------|------|------|------|------|------|
| AAC | 0.578 | 0.482 | 0.461 | 0.598 | 0.482 | 0.506 | 0.566 | 0.603 |
| DPC | 0.633 | 0.529 | 0.480 | 0.607 | 0.555 | 0.592 | 0.545 | 0.639 |
| CTD | 0.596 | 0.411 | 0.436 | 0.585 | 0.433 | 0.433 | 0.508 | 0.579 |
| GAAC | 0.577 | 0.465 | 0.482 | 0.600 | 0.478 | 0.502 | 0.550 | 0.615 |
| GDPC | 0.629 | 0.535 | 0.498 | 0.606 | 0.510 | 0.577 | 0.546 | 0.626 |
| GTPC | 0.519 | 0.494 | 0.470 | 0.515 | 0.478 | 0.475 | 0.472 | 0.516 |
| 8M_ESM2 | **1.000** | 0.998 | 0.994 | **0.999** | 0.985 | **0.992** | **0.976** | **1.000** |
| 35M_ESM2 | 0.999 | **0.999** | **0.996** | 0.997 | **0.987** | 0.991 | 0.971 | 0.999 |

**Note:**
The highest values are highlighted in bold.

a plant miPEPs prediction model. To optimize the performance of each method, an exhaustive hyperparameter search was performed to determine the most suitable hyperparameters for each machine learning method. The results of these models on the training set are presented in Table 4.

The three best performing models on both scales of ESM2 were LR, MLP and SVM, with the model utilizing the 35M parameter scale of ESM2 (35M_ESM2) performing slightly better than the 8M parameter scale model (8M_ESM2). We further evaluate their performance on independent test sets. The AUROC of the model using 35M_ESM2 to extract peptide embeddings, coupled with the aforementioned three ML methods as classifiers, is shown in Fig. 3.

The models with LR and MLP as classifiers perform significantly better than the models with SVM as classifier. Combined with the performance of the prediction models on the training set, we select LR to construct the final prediction model.

## Model performance comparison with state-of-the-art (SOTA) models

To assess the performance of our developed model pLM4PEP, we conducted a comparative analysis with the miPEPPred-FRL model developed by *Li et al. (2023)*. This comparison was performed across three distinct independent test sets, each representing a unique plant species. The evaluation metrics used in this assessment included AUROC and AUPRC, which are visually displayed in Table 5. Our model, pLM4PEP, demonstrated a marked advantage over the miPEPPred-FRL model across all three independent testing sets. This significant outperformance underscores the potential of pLM4PEP as a robust and reliable tool for plant miPEP prediction.

## Model performance across various bioactive peptides datasets

To further validate the generalization ability of our model, we assessed its performance on a diverse set of bioactive peptide datasets. Specifically, we selected three bioactive peptide datasets: the neuropeptide dataset, the blood-brain barrier peptide dataset, and the anti-parasitic peptide dataset.

PredNeuroP (*Bin et al., 2020*) is a neuropeptide prediction model based on a two-layer stacking method. It utilizes a combination of nine feature descriptors and five machine

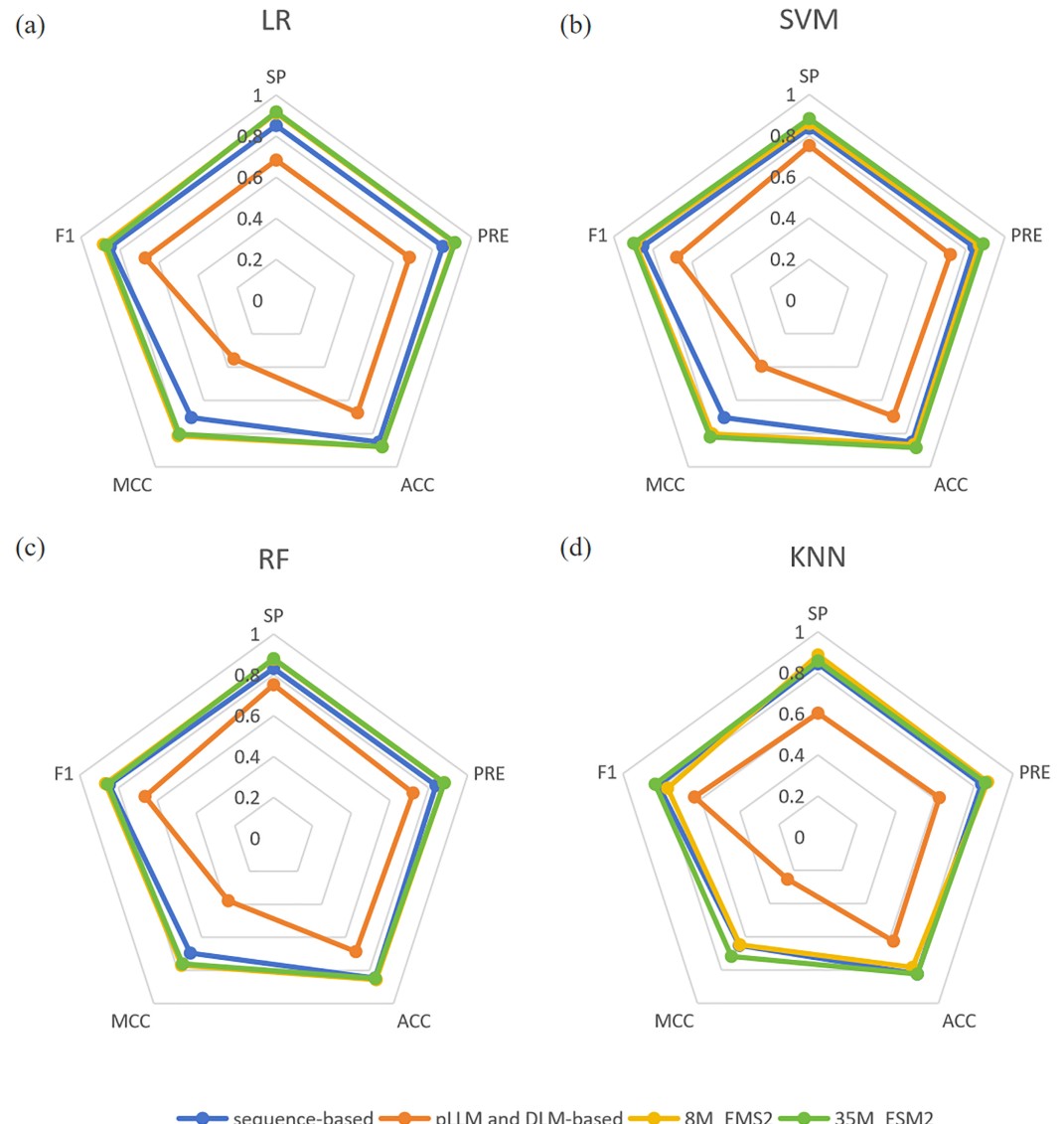

Figure 2 (A–D) Comparison of four feature extraction methods. "sequence-based" means the conventional protein sequence feature descriptors, "pLLM and DLM-based" means the protein large language model and deep learning model feature descriptors.

learning algorithms to generate 45 models as base learners. From these, eight base learners are selected for the first-layer learning. The outputs of these eight base learners are then fed into a logistic regression classifier to train the final model, and the output is the final prediction result, which constitutes the second-layer learning. BBPpred (*Dai et al., 2021*) is a blood-brain barrier peptide prediction model constructed based on a feature pool consisting of 103 feature encoding symbols derived from 16 feature encoding strategies. A three-step feature selection method is employed to select the optimal seven-dimensional features. The logistic regression method is used as the classifier in BBPpred. PredAPP (*Zhang et al., 2022*) is an anti-parasitic peptide prediction model. It generates 54 classifiers

**Table 4 Model performance of ESM2 at two scales combining different ML methods.**

| Method | ESM2 | Specificity | ACC | Precision | F1 | MCC | AUROC |
|---|---|---|---|---|---|---|---|
| LR | 8M | 0.912 | 0.878 | 0.911 | 0.885 | 0.813 | **1.000** |
| | 35M | **0.918** | 0.878 | **0.915** | 0.871 | 0.802 | 0.999 |
| MLP | 8M | 0.903 | 0.868 | 0.902 | 0.881 | 0.803 | **1.000** |
| | 35M | 0.910 | 0.874 | 0.908 | 0.872 | 0.798 | 0.999 |
| SVM | 8M | 0.859 | 0.866 | 0.869 | 0.888 | 0.802 | 0.999 |
| | 35M | 0.883 | **0.884** | 0.889 | **0.898** | **0.820** | 0.997 |
| AdaBoost | 8M | 0.899 | 0.870 | 0.899 | 0.880 | 0.795 | 0.998 |
| | 35M | 0.887 | 0.860 | 0.887 | 0.867 | 0.775 | 0.999 |
| XGBoost | 8M | 0.884 | 0.864 | 0.886 | 0.873 | 0.780 | 0.994 |
| | 35M | 0.908 | 0.868 | 0.906 | 0.874 | 0.796 | 0.996 |
| ERT | 8M | 0.875 | 0.858 | 0.878 | 0.869 | 0.772 | 0.992 |
| | 35M | 0.875 | 0.856 | 0.878 | 0.867 | 0.775 | 0.991 |
| RF | 8M | 0.877 | 0.856 | 0.879 | 0.867 | 0.769 | 0.985 |
| | 35M | 0.880 | 0.848 | 0.880 | 0.856 | 0.760 | 0.987 |
| KNN | 8M | 0.887 | 0.784 | 0.870 | 0.771 | 0.648 | 0.976 |
| | 35M | 0.858 | 0.824 | 0.858 | 0.835 | 0.718 | 0.971 |

**Note:**
The highest values are highlighted in bold.

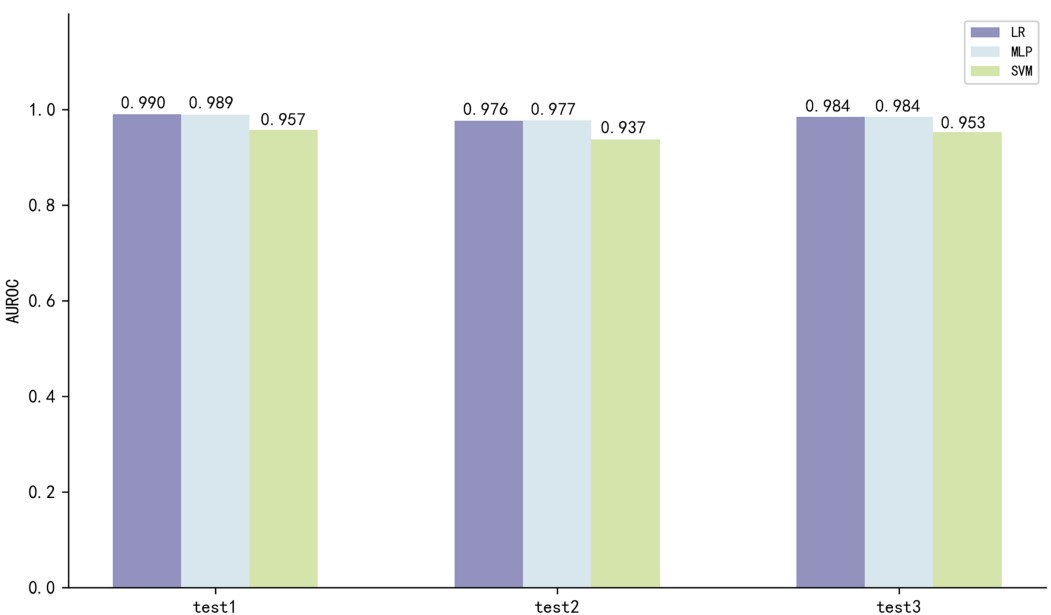

**Figure 3 Model performance of 35M_ESM2 on the independent testing datasets.**

**Table 5 Performance comparison with the original model on the independent testing datasets.**

| Method | miPEPPred-FRL | | pLM4PEP | |
|---|---|---|---|---|
| | AUROC | AUPRC | AUROC | AUPRC |
| Independent testing dataset 1 | 0.898 | 0.466 | 0.990 | 0.925 |
| Independent testing dataset 2 | 0.813 | 0.357 | 0.976 | 0.867 |
| Independent testing dataset 3 | 0.811 | 0.173 | 0.984 | 0.613 |

based on nine feature sets and six machine learning algorithms, resulting in 54 feature representations. Within each feature set, the best-performing feature representation is selected, and these selected feature representations are integrated using the logistic regression algorithm to construct the final model for anti-parasitic peptide prediction.

In the field of peptide prediction, feature selection plays a crucial role as it facilitates the identification of the most informative features associated with bioactive peptides from a large feature pool. These feature descriptors, including sequence, structure, and function features, encompassing crucial information from multiple aspects of bioactive peptides. The process of constructing these prediction models takes into consideration feature selection and algorithmic combinations from multiple perspectives, with the goal of enhancing prediction performance and improving generalization capability. By integrating multiple feature descriptors and employing machine learning algorithms, these models are able to capture the key features of a wide range of bioactive peptides from different angles, thus leading to improved accuracy and reliability in predictions.

It is noteworthy that in current research, the improvement of model prediction performance often relies on increasing the complexity of model development, involving intricate feature selection methods and modeling strategies. However, the use of ESM2 streamlines this process by eliminating the need for additional prepossessing steps prior to modeling, thus simplifying model development (*Du et al., 2024*).

UniDL4BioPep (*Du et al., 2023*) is a versatile deep learning architecture designed for binary classification of peptide bioactivity. It is built upon the ESM2 peptide embedding and CNN model. It has shown excellent performance on multiple bioactive peptide datasets. We compared our model pLM4PEP, with UniDL4BioPep, and those constructed in the original literature on three selected bioactive peptide datasets. The performance results of these models on the testing set are presented in Table 6 and Table S2.

It can be observed that pLM4PEP achieve superior results across all three bioactive peptide datasets when compared to the original models. Notably, our models exhibit particularly impressive performance on the neuropeptide and antiparasitic peptide datasets, surpassing the outcomes of UniDL4BioPep. However, it is observed that our models record a slight decrease in performance on the blood-brain barrier peptide dataset relative to UniDL4BioPep. Overall, our pLM4PEP model presents outstanding performance, reinforcing its effectiveness as a robust predictive tool. Both pLM4PEP and UniDL4BioPep utilize ESM2 to extract peptide feature embeddings. The primary reason for the performance variation between the models may be the disparities in classifier

**Table 6 Performance comparison with SOTA models on bioactive peptides datasets.**

| Dataset | Model | AUROC |
| --- | --- | --- |
| Neuropeptides dataset | PredNeuroP | 0.954 |
| | UniDL4BioPep | 0.953 |
| | pLM4PEP | **0.993** |
| Blood-brain barrier peptides dataset | BBPpred | 0.928 |
| | UniDL4BioPep | **0.992** |
| | pLM4PEP | 0.981 |
| Anti-parasitic peptides dataset | PredAPP | 0.922 |
| | UniDL4BioPep | 0.940 |
| | pLM4PEP | **0.994** |

**Note:**
The highest values are highlighted in bold.

selection. Multiple studies have already showcased the effectiveness and compatibility of LR in conjunction with ESM2 (*Chandra et al., 2023*; *Du et al., 2024*). Moreover, CNN necessitates a substantial volume of data to mitigate overfitting. Due to the limited size of the dataset for bioactive peptides, LR is likely to exhibit better performance. Furthermore, CNN demonstrates spatial translation invariance, thereby failing to leverage the potential positional information within the input sequences. Thus, despite CNN's notable accomplishments in image data and other domains, its performance in sequence modeling is comparatively less successful (*Chandra et al., 2023*). These results also underscore the advantages of employing ESM2 for feature embedding, showcasing its capability to capture the nuances of bioactive peptides for enhanced predictive accuracy.

Furthermore, we selected additional plant peptide datasets for experimentation. PTPAMP (*Jaiswal, Singh & Kumar, 2023*) is a prediction tool for plant-derived peptides and includes four types of peptide datasets: Antimicrobial, Antibacterial, Antifungal, and Antiviral. We conducted experiments on these four types of plant-derived peptide datasets, and the results are presented in Table S3. The results indicate that while our model demonstrates a certain degree of generalization capability, further research is needed in the future to enhance the model's applicability to a wider range of plant species peptides.

## Case study

To further verify the effectiveness of our method, pLM4PEP for identifying miPEPs, we performed a case study. We randomly selected three peptides for this purpose. The first example is an miPEP with experimental validation (*Su et al., 2021*), while the latter two are derived from non-miPEP sequences. Specifically, they were pseudo-peptide sequences translated from small nucleolar RNA (*Li et al., 2023*) in *Oryza sativa* and *Solanum lycopersicum* using small open reading frames. This selection enabled us to assess the method's discriminatory capacity between miPEPs and non-miPEP counterparts. The detailed prediction results are presented in Table 7.

In our analysis, the first peptide sample, an established miPEP, was correctly identified by our model with a prediction probability of 1.000, whereas the traditional manual

**Table 7 Detailed prediction results of miPEPPred-FRL and pLM4PEP on the three case study peptides.**

| Peptide type | Sequence | miPEPPred-FRL | pLM4PEP |
|---|---|---|---|
| miPEP | MVYIFQLVLISRLGL | 0.379 | 1.000 |
| non-miPEP | MMNNSSQISFMDLFSE | 0.736 | 0.075 |
| non-miPEP | MKMMNIRRNDDQFLFK | 0.828 | 0.026 |

feature-based model, miPEPPred-FRL, assigned a significantly lower probability of 0.379, incorrectly classifying the sequence as non-miPEP. For the subsequent non-miPEP samples, our model predicts them as miPEP with much lower probabilities of 0.075 and 0.026, respectively, thereby correctly categorizing them as non-miPEP. Conversely, miPEPPred-FRL predicts these samples with probabilities of 0.736 and 0.828, respectively, and thus incorrectly suggesting these sequences as miPEPs This comparison underscores the superior discriminative ability of our model in classifying miPEPs.

## CONCLUSIONS

In this study, we developed a predictive model named pLM4PEP, which integrates ESM2 peptide embedding and machine learning techniques. Experimental results indicate that the pLM4PEP model excels in performance across four distinct datasets: plant miPEPs, neuropeptides, blood-brain barrier peptides, and anti-parasitic peptides. This suggests that pLM4PEP has the potential to be a universally applicable method for peptide prediction, capable of addressing various challenges in peptide identification across different biological contexts. However, it is important to note that despite these promising results, we have not yet undertaken biological experiments to validate our predictive findings. The lack of empirical validation represents a gap that needs to be addressed. Rigorous experimental validation will strengthen the reliability and practicality of our predictions in real-world scenarios. In addition to evaluating the model's performance, we also provided an in-depth examination of critical factors that influence the construction of an effective predictive model. We explored how different features and machine learning algorithms impact the model's performance, enabling a thorough comparative analyses. This research also highlights the outstanding performance of ESM2 as a feature embedding method, which can greatly improve the accuracy and efficacy of miPEP identification. We anticipate that this study will not only accelerate the discovery and research of plant miPEPs but also provide support for the identification and functional analysis of a wide range of other bioactive peptides. In the future, we will continue to enhance the applicability of the model to a broader range of plant species and miPEP types, as well as to improve its generalization performance for other bioactive peptides, thereby making contributions to the field of peptide research.

## Funding

This work was supported by the National Natural Science Foundation of China (62362062), Autonomous Region "Tianshan Talents" Young Top Talents-Young Scientific and Technological Innovation Talents (2023TSYCCX0104), Multimodal Major Chronic Disease Prevention and Control Science and Engineering Research Project (No.MCD -2023-1-15) and the Natural Science Foundation of the Anhui Higher Education Institutions of China (2023AH051392). The funders had no role in study design, data collection and analysis, decision to publish, or preparation of the manuscript.

## Grant Disclosures

The following grant information was disclosed by the authors:
National Natural Science Foundation of China: 62362062.
Autonomous Region "Tianshan Talents" Young Top Talents-Young Scientific and Technological Innovation Talents: 2023TSYCCX0104.
Multimodal Major Chronic Disease Prevention and Control Science and Engineering Research Project: MCD -2023-1-15.
Natural Science Foundation of the Anhui Higher Education Institutions of China: 2023AH051392.

## Competing Interests

The authors declare that they have no competing interests.

## Author Contributions

- Yishan Yue conceived and designed the experiments, performed the experiments, analyzed the data, performed the computation work, prepared figures and/or tables, authored or reviewed drafts of the article, and approved the final draft.
- Henghui Fan conceived and designed the experiments, performed the computation work, authored or reviewed drafts of the article, and approved the final draft.
- Jianping Zhao conceived and designed the experiments, authored or reviewed drafts of the article, and approved the final draft.
- Junfeng Xia conceived and designed the experiments, authored or reviewed drafts of the article, and approved the final draft.

## Data Availability

    The code is available at GitHub: https://github.com/xialab-ahu/pLM4PEP.
    The plant miPEPs dataset is available at GitHub: https://github.com/HiBLee/miPEPPred-FRL.
    The neuropeptides dataset is available at GitHub: https://github.com/xialab-ahu/PredNeuroP.
    The blood-brain barrier peptides dataset is available at: http://BBPpred.xialab.info.

The anti-parasitic peptides dataset is available at GitHub: https://github.com/xialab-ahu/PredAPP.git.

The datasets of PTPAMP is available at: http://www.nipgr.ac.in/PTPAMP.

The datasets and code are also available at Zenodo: Yue, Y. (2025). Code and related datasets for pLM4PEP. Zenodo. https://doi.org/10.5281/zenodo.14829973.

## Supplemental Information

Supplemental information for this article can be found online at http://dx.doi.org/10.7717/peerj-cs.2733#supplemental-information.

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
