# Peer review of "Protein language model-based prediction for plant miRNA encoded peptides"

_PeerJ Computer Science, doi:10.7717/peerj-cs.2733_

## Round 0.1 · original submission · Major Revisions

Please revise the article carefully according to the comments. Your work will be evaluated again.

Reviewer 1 ·

Basic reporting

In this manuscript, the authors introduce a computational model designed to predict plant miRNA-encoded peptides (miPEPs).

Experimental design

This model integrates pre-trained language model embeddings with machine learning techniques. The primary objective is to harness transfer learning from extensive protein datasets, capturing the subtleties within peptide sequences without the need for manually crafted features.

Validity of the findings

Remarkably, the pLM4PEP model outperforms specialized tools in predicting neuropeptides, blood-brain barrier peptides, and anti-parasitic peptides, suggesting its potential as a versatile tool for diverse peptide prediction tasks.

Additional comments

1. The use of protein language models for feature embedding is highly effective. A comparison with other tools employing ESM2 or similar protein language models could bolster the credibility and practical application of this model.
2. Given that the pLM4PEP model has shown efficacy across various bioactive peptide datasets beyond miPEPs, the authors might consider rebranding this tool as a universal peptide prediction method, with the prediction of plant miPEPs as just one of its many applications.
3. As the manuscript encompasses a range of bioactive peptide datasets, not just plant miPEPs, it would be beneficial to include a concise introduction to bioactive peptides to provide context. Additionally, incorporating recent literature could enrich the discussion on this topic.
4. The authors might elaborate on the rationale behind choosing specific traditional manual feature encodings for comparison, to clarify their methodology and decision-making process.
5. It would be prudent for the authors to discuss the model's limitations and potential areas for future enhancement and research in the conclusion section.
6. The manuscript's language presentation could be refined in several areas:
6.1 Line 53: The phrase "this need" should be specified to better articulate the research motivation and its importance.
6.2 Line 198: The phrase "ESM2 feature embedding analysis" could be rephrased to "The analysis of ESM2 feature embedding" for improved grammatical structure.
6.3 Lines 210-211: The term "this purpose" should be replaced with a more explicit expression to accurately convey the intended meaning in this context.

Cite this review as

·

Basic reporting

Overall language usage is good, although there are some typos(e.g. line 58).
The first part of the Results and Discussion is how the handcrafted features are designed, followed by the list of the different predictive models implemented. I believe these parts would better fit in the Method section, and later can be expanded in the Results section where appropriate. There are also other sections in the the results which could be more appropriate as conclusion, such as parts in the "Model performance across various bioactive peptides datasets".
Certain parts such as Methods and Conclusions can be expanded upon to facilitate the understanding of the research done.

Experimental design

The study focuses on the devolpment of a plant miPEPs predictor tool. While the performance across the different datasets(neuropeptide, blood-brain barrier and anti-parasitic) is good, indicating generalization of the model, it is not explained why these particular datasets were chosen. Peptide classes are numerous and there are plant peptide databases that could have been used in the analysis; on the other hand, if the authors intended to create a more generalizable model, why not use a more comprehensive database for training the model? Please elaborate.

Which hyperparameters did you use during the optimization of the machine learning model performance? I would suggest adding a brief explanation for each model, either in the text or in a Supplementary table.

Validity of the findings

The comparison of the tool with UniDL4BioPep only reports the AUROC values on the three independent dataset. The other metrics should also be included in the comparison to better understand the impact of the different classifiers.
In the case study, there is no mention of the sources from where the peptides are taken from. If the authors wanted to compare the differences in predicting miPEPs and non-miPEP they should provide a more comprehensive criterion on why these peptides were chosen and/or possibly extend the list of interested peptides.

Reviewer 3 ·

Basic reporting

This paper is clear and the references are correct.

Experimental design

The design is reasonable.

Validity of the findings

The findings need be validated via experiments.

Additional comments

In this paper, the authors present a predictive model named pLM4PEP, which integrates Evolutionary Scale Modeling (ESM2) peptide embeddings with machine learning techniques to identify plant microRNA-encoded peptides (miPEPs). The following lists some comments. First, the predictive findings have not been validated. The authors need provide some computational validations. Experimental validation will reinforce the reliability and applicability of the predictions in biological contexts. Second, the model exhibits excellent performance, it suggest further exploration into the integration of ESM2 with different machine learning algorithms to optimize peptide feature embeddings and enhance prediction accuracy. Third, the study also highlights the need for continued research to improve the model's generalization capabilities and its applicability to a broader range of plant species and miPEP types.

Cite this review as

---

## Round 0.2 · accepted · Accept

Thanks for your efforts to improve the work. This version addressed the concerns of the reviewers successfully. It may be accepted. Congrats!

Reviewer 1 ·

Basic reporting

The paper can be accepted.

Experimental design

The paper can be accepted.

Validity of the findings

The paper can be accepted.

Additional comments

The paper can be accepted.

Cite this review as

·

Basic reporting

Language is now fine, corrections were made and the text is clearer to read. Literature has been provided where requested and the structure of the various sections is now more defined.

Experimental design

Details were added to describe the focus and aim of the research with more clarity.

Validity of the findings

Including extra metrics for the models gives more robustness to the performance.

Reviewer 3 ·

Basic reporting

This paper proposed a protein language model-based prediction for plant miRNA encoded peptides.

Experimental design

This design of experiments is okay in the revised version.

Validity of the findings

The method has been validated.

Additional comments

The authors have addressed my former comments.

Cite this review as